# The evaluating prescription opioid changes in veterans (EPOCH) study: Design, survey response, and baseline characteristics

Erin E. Krebs[1,2]*, Barbara Clothier[1], Sean Nugent[1], Agnes C. Jensen[1], Brian C. Martinson[1,2,3], Elizabeth S. Goldsmith[1,4], Melvin T. Donaldson[4,5], Joseph W. Frank[6,7], Indulis Rutks[1], Siamak Noorbaloochi[1,2]

1 Center for Care Delivery and Outcomes Research, Minneapolis VA Health Care System, Minneapolis, MN, United States of America, 2 Department of Medicine, University of Minnesota Medical School, Minneapolis, Minnesota, United States of America, 3 HealthPartners Institute, Bloomington, MN, United States of America, 4 Division of Epidemiology and Community Health, University of Minnesota School of Public Health, Minneapolis, MN, United States of America, 5 Medical Scientist Training Program, University of Minnesota Medical School, Minneapolis, MN, United States of America, 6 Center of Innovation for Veteran-Centered and Value-Driven Care, VA Eastern Colorado Health Care System, Aurora, Colorado, United States of America, 7 Division of General Internal Medicine, University of Colorado School of Medicine, Aurora, CO, United States of America

* erin.krebs@va.gov

**Data Availability Statement:** Data cannot be shared publicly because they are sensitive human research participant data and their release would be inconsistent with the study's Institutional Review

## Abstract

In the United States (US), long-term opioid therapy has been commonly prescribed for chronic pain. Since recognition of the opioid overdose epidemic, clinical practice guidelines have recommended tapering long-term opioids to reduced doses or discontinuation. The Effects of Prescription Opioid Changes for veterans (EPOCH) study is a national population-based prospective observational study of US Veterans Health Administration primary care patients designed to assess effects of evolving opioid prescribing practice on patients treated with long-term opioids for chronic pain. A stratified random sampling design was used to identify a survey sample from the target population of patients treated with opioid analgesics for $\geq$ 6 months. Demographic, diagnostic, visit, and pharmacy dispensing data were extracted from existing datasets. A 2016 mixed-mode mail and telephone survey collected patient-reported data, including the main patient-reported outcomes of pain-related function (Brief Pain Inventory interference; BPI-I scores 0–10, higher scores = worse) and health-related quality of life. Data on survey participants and non-participants were analyzed to assess potential nonresponse bias. Weights were used to account for design. Linear regression models were used to assess cross-sectional associations of opioid treatment with patient-reported measures. Of 14,160 patients contacted, 9253 (65.4%) completed the survey. Participants were older than non-participants (63.9 ± 10.6 vs. 59.6 ± 13.0 years). The mean number of bothersome pain locations was 6.8 (SE 0.04). Effectiveness of pain treatment and quality of pain care were rated fair or poor by 56.1% and 45.3%, respectively. The opioid daily dosage range was 1.6 to 1038.2 mg, with mean = 50.6 mg (SE 1.1) and median = 30.9 mg (IQR 40.7). Among the 73.2% of patients who did not receive long-acting opioids, the mean daily dosage was 30.4 mg (SE 0.6) and mean BPI-I was 6.4 (SE 00.4). Among patients who received long-acting opioids, the mean daily dosage was 106.2 mg

Board (IRB) approval and United States Department of Veterans Affairs (VA) privacy, confidentiality, and information security regulations. Data are available for researchers who meet the criteria for access to sensitive VA data (contact IRBMN@VA.GOV or study authors for information).

**Funding:** The study was supported by the United States Department of Veterans Affairs, Veterans Health Administration, Health Services Research and Development (IIR 14-295 to EEK and CDA 15-059 to JWF) and National Institutes of Health (F30AT009162 to MTD). The funders had no role in study design, data collection and analysis, decision to publish, or preparation of the manuscript.

**Competing interests:** The authors have declared that no competing interests exist.

(SE 2.8) and mean BPI-I was 6.8 (SE 0.07). Higher daily dosage was associated with worse pain-related function and quality of life among patients without long-acting opioids, but not among patients with long-acting opioids. Future analyses will use follow-up data to examine effects of opioid dose reduction and discontinuation on patient outcomes.

## Introduction

In the United States (US), opioids are commonly prescribed for long-term management of chronic pain. In 2013–2014, 5.4% of US adults reported taking opioids for >90 days.[1] Similarly, in 2016, 6.2% of US Veterans Health Administration (VA) patients were dispensed opioid therapy for >90 consecutive days.[2] Opioid prescribing for chronic pain has usually been open-ended, often resulting in opioid use for many years.

Recent studies show longer duration of opioid therapy is associated with greater risk of serious harms, including death. Compared with patients prescribed short-term or intermittent opioids, those prescribed long-term opioid therapy (LTOT) are more likely to receive long-acting opioids and higher opioid daily doses, treatment factors that are also associated with greater risk of serious harms.[3–6]

In response to growing evidence of opioid-related harms, the US Centers for Disease Control and Prevention (CDC) and VA/Department of Defense (DoD) released opioid prescribing guidelines that recommend limiting the frequency, intensity, and duration of opioid therapy for chronic pain.[7, 8] For patients already prescribed LTOT, the guidelines recommend tapering to reduced doses or discontinuation when benefits do not clearly outweigh potential harms. Although some studies have found improvements in pain and quality of life with opioid dose reduction, only very low-quality evidence is available to guide opioid tapering practice. Research is needed to understand outcomes of opioid dose reduction and discontinuation in practice.

The Effects of Prescription Opioid Changes for veterans (EPOCH) study is a nationwide prospective population-based observational study of US VA primary care patients treated with LTOT. The primary study objective is to evaluate patient-reported outcomes of changes in opioid prescribing, especially opioid dose reduction and discontinuation, in an era of rapidly changing opioid prescribing practice. The purpose of this report is to describe EPOCH study methods, survey response, and baseline clinical and opioid treatment characteristics, including associations of opioid treatment factors with patient-reported outcome measures.

## Methods

The Minneapolis VA Health Care System Institutional Review Board approved the study, including a waiver of written informed consent. The target population was VA primary care patients treated with LTOT for chronic pain. A two-stage stratified random sampling design was used to select a survey sample of eligible patients. Eligible patients were selected at random from among panels of primary care providers (PCPs)_ who met minimum panel size criteria who were selected at random from among all US-based VA facilities. A multiple-contact (mail and telephone) tailored design approach was used to collect patient-reported data.

## Eligibility criteria

Eligible patients had current LTOT, at least one primary care clinic visit within 12 months before the most recent opioid dispensing date, and no indication for opioid therapy other than chronic pain. Current LTOT was defined as 1) a qualifying opioid analgesic dispensed within the prior 30 days and 2) ≥150 days' supply of a qualifying opioid in the 180 days before the most recent dispensing date with no between-fill gaps >40 days. Qualifying opioid analgesics were on the VA formulary and indicated for pain, not including tramadol or buprenorphine (S1 Table). Patients were excluded if data indicated a likely indication for LTOT other than chronic pain, such as active cancer treatment, end of life care, or opioid use disorder (S2 Table).

## Administrative and electronic medical record data

Patients were identified and contacted in monthly waves. Each month, updated data were extracted from the VA Corporate Data Warehouse (CDW) and patients were randomly selected from among those eligible that month. Selected patients were invited to participate. For each selected patient, the index date was defined as the most recent opioid dispensing date before selection. For patients who were eligible in more than one extraction month and not selected for the survey sample, the index date was randomly chosen from among their possible index dates. Demographic and clinical variables were extracted from the VA CDW for the year prior to the index date. The Charlson comorbidity index was used as a measure of medical morbidity.[9] Clinically relevant pain and mental health diagnosis categories were created based on work of other VA researchers (S3 Table).

Opioid receipt was determined from VA outpatient pharmacy dispensing data. Morphine-equivalent (ME) opioid daily dosage was calculated using CDC-recommended conversion factors (S1 Table).[10] Daily dosage was calculated by summing all opioids dispensed in the six months before and including the index date prescription and dividing by the number of days from the first opioid dispensed in the prior 6 months to the end of the index prescription days' supply. Opioid formulations were categorized by duration of effect as long-acting or short-acting. Patients were categorized according to whether they received any long-acting opioid or no long-acting opioid (i.e., only short-acting opioids) in the six months before and including the index prescription. For descriptive purposes, opioid daily dosage was categorized with conventional cutoffs as low (<20 mg), moderate (20 to <50 mg), high (50 to <100 mg), or very high (≥100 mg). Analyses treated daily dosage as a continuous variable in 10 ME mg increments.

## Survey sample selection

A stratified random sampling design was used to identify eligible patients for the survey sample from among primary care panels at 140 VA parent health care systems. Each of these health care systems is an administrative unit that comprises one or more hospitals and their affiliated outpatient community-based clinics. All 140 US-based VA health care systems were included; a VA facility located in the Philippines was not included.

Primary care providers (PCP) are defined by VA as physicians, advanced practice nurses, or physician assistants who provide primary care to an assigned panel of patients. PCPs and their assigned patients at each health care system were identified using CDW data from the primary care management module. To ensure adequately sized clusters for analysis, minimum PCP panel size criteria were applied. These criteria were a) at least 500 total patients assigned to the panel and b) at least 4 patients receiving long-term opioid therapy assigned to the panel.

At the time of the first data extraction, all PCPs who met panel size criteria were identified and a list of these PCPs was created for each health care system. New PCPs who were identified

with subsequent monthly extractions were added to their health care system's provider list. To select patients for the survey panel, we first selected a random sample of PCPs from the provider list of each VA health care systems. Subsequently, a simple random sample of eligible patients was selected from each selected PCP's panel.

## Survey data collection

Scannable paper questionnaires were designed using Teleform software. A draft questionnaire was tested in cognitive interviews with six patients and an embedded pilot was conducted by completing an initial survey wave with 500 patients (April-May 2016). After minor adjustments to the questionnaire and survey procedures, six more survey waves were conducted at monthly intervals beginning in June 2016. Responses were accepted until April 2017.

A multiple-contact (mail and telephone) tailored design approach was used for data collection.[11] Patients were mailed an initial letter with study information and instructions for opting out. This was followed by a survey packet including a cover letter, scannable paper questionnaire, and postage-paid return envelope. Patients who did not respond were contacted with a mailed reminder, followed by a second round of mailed survey packets and reminders, then telephone calls. Patients reached by telephone were offered telephone interviews. Five-dollar checks were mailed after responses were received.

Each completed questionnaire was reviewed for completeness and comments, then scanned twice, with verification by two research associates per document. Consistency checks were performed for out of range values and missing data. For multi-item measures, a score was calculated if at least 2/3 of items were complete; otherwise the measure was considered missing.

## Main patient-reported measures

The main outcome measures were pain-related function and health-related quality of life. Pain-related function was assessed with the 7-item Brief Pain Inventory Interference (BPI-I) scale, which includes seven 0–10 numeric ratings of pain interference with general activity, mood, walking, work, relations with other people, sleep, and enjoyment of life and is scored as the average of individual item scores (0–10 score, higher = worse).[12, 13] Health-related quality of life was assessed with the Veterans RAND 12-item Health Survey (VR-12), a measure adapted from the 36-item Medical Outcomes Study health survey (SF-36) that includes self-rated health "in general, how would you rate your health?" (excellent, very good, good, fair, poor)[14] and 11 other items used to calculate physical and mental summary scores (0–100 scores; standardized and normed to the US population mean of 50 and SD of 10; higher = better).[15, 16]

## Additional measures

Secondary patient-reported measures included pain severity, pain location, satisfaction with pain care, and preferences for treatment. Pain was characterized with a numeric rating of average pain severity over the past week (0–10 score, higher = worse) and by asking about presence of bothersome pain in the past 6 months (response options: not bothered at all, bothered a little, or bothered a lot) at the following locations: headache; teeth, mouth, or jaw; neck; back; shoulder; hip; knee; foot, ankle, or lower leg; stomach or abdomen; pelvis or genitals; widespread pain all over your body.[17] Teeth, mouth, or jaw and foot, ankle, or lower leg were not asked on the pilot questionnaire. Satisfaction with pain care was assessed with two questions rating 1) "overall effectiveness of your pain treatment" and 2) "quality of pain care you received from the VA in the past 12 months" (response options: poor, fair, good, very good, excellent). To assess preference for opioid treatment, patients were asked to rate agreement

with statements about past-year desire for "my doctor to prescribe stronger or higher dose opioid medicines" and "to stop using opioid medicines or cut down on the amount of opioid medicines" (response options strongly disagree, disagree, neutral, agree, strongly agree).[18]

## Statistical analysis

To assess survey nonresponse mechanisms and potential bias, we used logistic regression to model participation (i.e., comparing those who completed a questionnaire or interview with those who were invited but unreachable, refused, or did not respond) on 24 prespecified variables of interest including variables related to pain, opioid dosage, and mental health or substance use diagnoses. Patients with complete data on all 24 variables (n = 13,976) were included in models. Initially, eleven variables were significant at the p<0.05 level; however, p-values often approach zero in large samples even when differences are not practically significant.[19, 20] To address this large sample problem, we investigated the robustness of significant differences by varying sample sizes. We used three sets of random seed generators to create twelve series of distinct and independent datasets of sizes 400 to 1000 (with increments of 200) and 1000 to 5000 (with increments of 1000). Separately in each dataset, we used the same logistic regression approach to identify predictors with sample size-robust statistical significance. Variables with p<0.05 only in sample sizes >2000 were treated as practically non-significant.

To account for survey design, design weights were calculated by taking the inverse of the probability of provider selection multiplied by the probability of patient selection.[21–23] All survey cohort analyses used weights to account for design (using parent health care system as the strata variable, provider as the primary unit, and patients within provider as the secondary unit) and adjusted for age to account for nonresponse. Provider clusters with only one selected patient (n = 171) were dropped from weighted analyses; therefore, results for the survey cohort are based on 9,074 participating patients.

Associations of opioid formulation (any long-acting versus no long-acting) and opioid daily dosage (continuous 10 mg increments or categorized in 4 groups) with patient-reported outcome measures (BPI-I, VR-12 physical, VR-12 mental) were tested using age-adjusted weighted linear regression models including patients with complete outcomes. Because long-acting opioids are available in higher dose units than short-acting opioids and distribution of daily dosages differed between patients who did and did not receive long-acting opioids, we examined whether receiving a long-acting formulation modified the association of dosage with outcomes by adding interaction terms for dosage by formulation to linear regression models. Some 10 mg dosage intervals had no or very few (i.e., <3) patients because few long-acting users were at the lowest end of the dose distribution and few non-long-acting users were at the highest end. After considering both dosage distributions and clinical relevance, we excluded patients with dosages <10 or >200 from models evaluating dose by long-acting interactions. Subsequent models were stratified by receipt of long-acting opioids. Sensitivity analyses excluded patients at the high end of the daily dosage range (≥500 for long-acting and ≥200 for no long-acting). Statistical significance was determined by p<0.05. The Survey R package was used for analyses.[23]

## Results

Fig 1 shows how eligible patients were selected for contact and enrolled in the survey cohort. Of 14,160 patients we attempted to contact, 9253 (65.4%) completed a questionnaire or interview and were enrolled as participants in the survey cohort. Of all enrolled participants, 732 (7.9%) completed the survey by telephone.

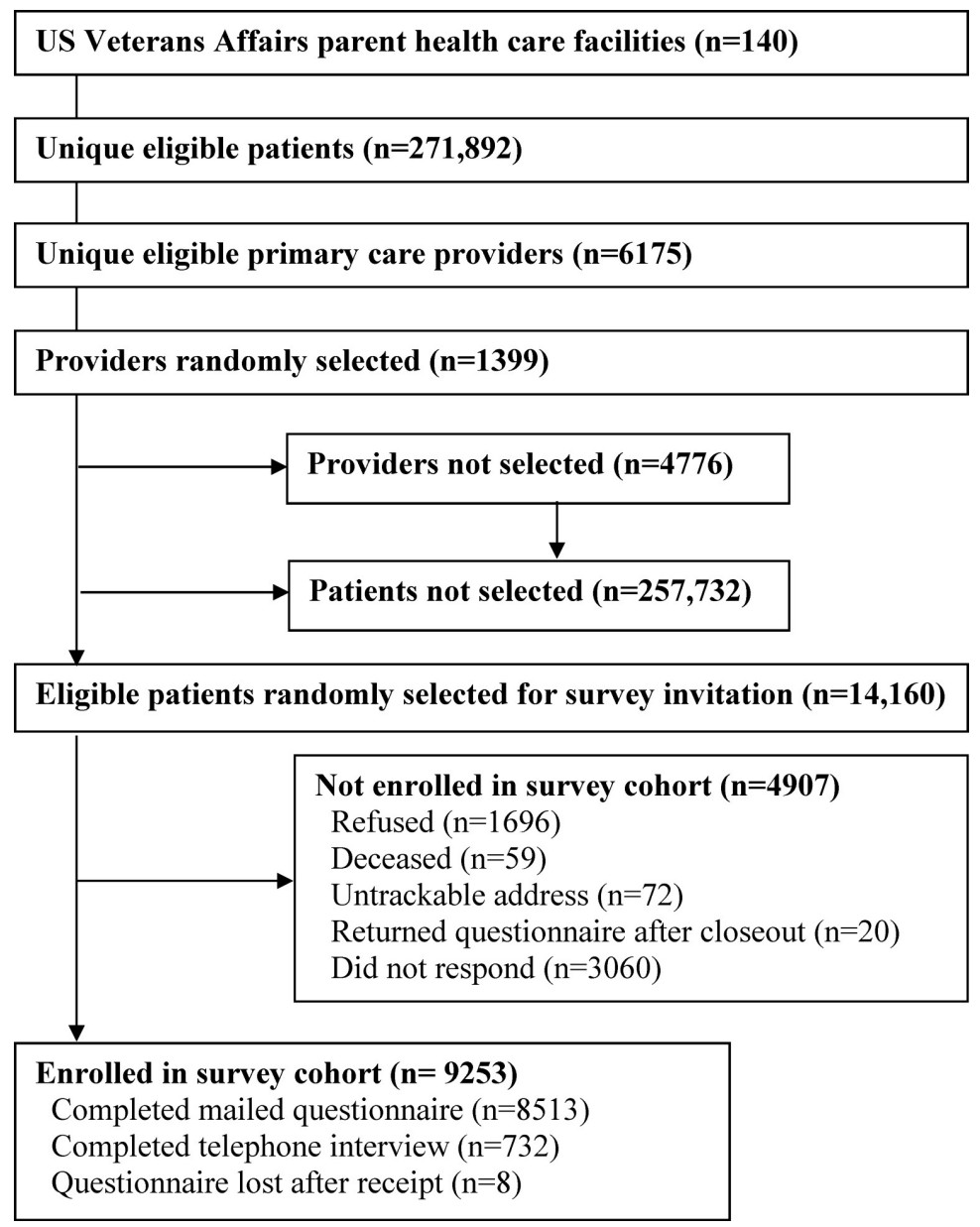

**Fig 1. Study flow diagram.**

Table 1 shows characteristics of eligible patients by participation status. Enrolled participants were 92.4% male, 79.3% white, 12.8% black, and 4.0% Hispanic. The most common pain-related diagnoses were back disorders (68.5%), osteoarthritis (30.2%) and neck disorders (21.2%). Participants frequently had mental health diagnoses, especially depressive disorders (29.8%) and PTSD (22.6%). Compared with patients who were selected but did not enroll, participants were older (63.9 ± 10.6 vs. 59.6 ± 13.0 years). In non-response bias analyses, age was the only predictor of study participation with robust statistical significance to sample sizes ≤ 2000 (S4 Table).

**Table 1. Characteristics of EPOCH-eligible patients by study participation status at index date.**

| | All eligible patients (n = 271,892) | Patients selected for invitation (n = 14,160) | Patients selected but not enrolled in survey cohort [a](n = 4907) | Participants enrolled in survey cohort (n = 9253) |
|---|---|---|---|---|
| Age in years | 62.4 (11.9) | 62.5 (11.7) | 59.6 (13.0) | 63.9 (10.6) |
| Sex, male | 252344 (92.8%) | 13022 (92.0%) | 4476 (91.2%) | 8546 (92.4%) |
| Race, | | | | |
| White | 209143 (76.9%) | 11145 (78.7%) | 3809 (77.6%) | 7336 (79.3%) |
| Black | 39333 (14.5%) | 1861 (13.1%) | 679 (13.8%) | 1182 (12.8%) |
| American Indian | 2829 (1.0%) | 134 (1.0%) | 49 (1.0%) | 85 (0.9%) |
| Asian | 718 (0.3%) | 47 (0.3%) | 21 (0.4%) | 26 (0.3%) |
| Pacific Islander | 1938 (0.7%) | 105 (0.7%) | 24 (0.5%) | 81 (0.9%) |
| Multi-race | 2637 (1.0%) | 147 (1.0%) | 51 (1.0%) | 96 (1.0%) |
| Unknown | 15294 (5.6%) | 721 (5.1%) | 274 (5.6%) | 447 (4.8%) |
| Hispanic ethnicity | | | | |
| Yes | 10489 (3.9%) | 613 (4.3%) | 247 (5.0%) | 366 (4.0%) |
| No | 251669 (92.6%) | 13095 (92.5%) | 4508 (91.9%) | 8587 (92.8%) |
| Unknown | 9734 (3.6%) | 452 (3.2%) | 152 (3.1%) | 300 (3.2%) |
| Married | 141637 (52.1%) | 7360 (52.0%) | 2474 (50.4%) | 4886 (52.8%) |
| Urban residence | 151827 (55.8%) | 8034 (56.7%) | 2906 (59.2%) | 5128 (55.4%) |
| VA enrollment priority group [c] | | | | |
| Service connected (SC; group 1–4) | 164422 (60.5%) | 8617 (61.1%) | 3074 (62.7%) | 5543 (60.1%) |
| Not SC, no copay (group 5–6) | 82519 (30.4%) | 4197 (29.8%) | 1386 (28.3%) | 2811 (30.5%) |
| Not SC, with copay (group 7–8) | 24184 (8.9%) | 1286 (9.1%) | 420 (8.6%) | 866 (9.4%) |
| Post-9/11 military service | 15943 (5.9%) | 799 (5.6%) | 442 (9.0%) | 357 (3.9%) |
| US census division | | | | |
| East North Central | 38066 (14.0%) | 2116 (14.9%) | 718 (14.6%) | 1398 (15.1%) |
| East South Central | 28470 (10.5%) | 1000 (7.1%) | 349 (7.1%) | 651 (7.0%) |
| Middle Atlantic | 15678 (5.8%) | 1828 (12.9%) | 668 (13.6%) | 1160 (12.5%) |
| Mountain | 31589 (11.6%) | 1403 (9.9%) | 452 (9.2%) | 951 (10.3%) |
| New England | 6235 (2.3%) | 804 (5.7%) | 292 (6.0%) | 512 (5.5%) |
| Pacific | 38143 (14.0%) | 1609 (11.4%) | 586 (11.9%) | 1023 (11.1%) |
| South Atlantic | 56615 (20.8%) | 2630 (18.6%) | 906 (18.5%) | 1724 (18.6%) |
| West North Central | 17767 (6.5%) | 1313 (9.3%) | 420 (8.6%) | 893 (9.7%) |
| West South Central | 3929 (14.5%) | 1457 (10.3%) | 516 (10.5%) | 941 (10.2%) |
| Average pain score in prior year [c] | 4.63 (2.38) | 4.62 (2.35) | 4.73 (2.38) | 4.56 (2.33) |
| Pain diagnoses [b] | | | | |
| Back/spine disorders | 178954 (65.8%) | 9609 (67.9%) | 3268 (66.6%) | 6341 (68.5%) |
| Neck/spine disorders | 53397 (19.6%) | 2970 (21.0%) | 1005 (20.5%) | 1965 (21.2%) |
| Osteoarthritis | 79161 (29.1%) | 4082 (28.8%) | 1284 (26.2%) | 2798 (30.2%) |
| Neuropathy | 50808 (18.7%) | 2668 (18.8%) | 800 (16.3%) | 1868 (20.2%) |
| Headache | 20906 (7.7%) | 1094 (7.7%) | 438 (8.9%) | 656 (7.1%) |
| Mental health diagnoses [b] | | | | |
| Depressive disorder | 74769 (27.5%) | 4122 (29.1%) | 1364 (27.8%) | 2758 (29.8%) |
| Anxiety disorder | 39457 (14.5%) | 2242 (15.8%) | 844 (17.2) | 1398 (15.1%) |
| PTSD | 62214 (22.9%) | 3282 (23.2%) | 1193 (24.3%) | 2089 (22.6%) |
| Alcohol use disorder | 17284 (6.4%) | 941 (6.7%) | 350 (7.1%) | 591 (6.4%) |

*(Continued)*

**Table 1.** (Continued)

|  | All eligible patients (n = 271,892) | Patients selected for invitation (n = 14,160) | Patients selected but not enrolled in survey cohort [a] (n = 4907) | Participants enrolled in survey cohort (n = 9253) |
|---|---|---|---|---|
| Drug use disorder | 16843 (6.2%) | 1015 (7.2%) | 379 (7.7%) | 636 (6.9%) |
| Charlson comorbidity score [b] | 1.43 (1.73) | 1.40 (1.70) | 1.27 (1.67) | 1.47 (1.71) |

Values are means and standard deviations (SD) or n and percent (%) except where indicated.

[a] Not included group comprises patients who refused, did not respond before study closeout, or were deceased or unreachable.

[b] From ICD-9 and ICD-10 diagnoses in the prior 12 months.

[c] Missing data: 767 (0.3%) unknown priority group treated as missing and 2313 (0.9%) missing pain scores.

Table 2 shows opioid treatment received by participation status. Most patients were treated with only short-acting opioids. The most common opioids received were hydrocodone (57.8%), oxycodone (34.8%), and morphine (19.4%).

## Survey cohort characteristics

Table 3 shows administrative, electronic medical record (EMR), and patient-reported measures for the survey cohort overall (n = 9074) and according to daily dosage category. The

**Table 2. Characteristics of EPOCH-eligible patients by study participation status at index date.**

|  | All eligible patients (n = 271,892) | Patients selected for invitation (n = 14,160) | Patients selected but not enrolled in survey cohort [a] (n = 4907) | Participants enrolled in survey cohort (n = 9253) |
|---|---|---|---|---|
| Opioid daily dose |  |  |  |  |
| ME mg/day, mean (SD) | 47.2 (59.8) | 51.9 (64.1) | 52.8 (65.7) | 51.5 (63.2) |
| ME mg/day, median (IQR) | 29.7 (33.7) | 30.4 (40.3) | 30.6 (40.6) | 30.4 (40.1) |
| Opioid formulation |  |  |  |  |
| Any long-acting | 63817 (23.5%) | 3964 (28.0%) | 1324 (27.0%) | 2640 (28.5%) |
| Short-acting only | 208075 (76.5%) | 10196 (72.0%) | 3583 (73.0%) | 6613 (74.5%) |
| Specific opioid dispensed |  |  |  |  |
| Hydrocodone | 171142 (62.9%) | 8091 (57.1%) | 2740 (55.8%) | 5351 (57.8%) |
| Oxycodone | 88131 (32.4%) | 5083 (35.9%) | 1863 (38.0%) | 3220 (34.8%) |
| Morphine | 44870 (16.5%) | 2685 (19.0%) | 890 (18.1%) | 1795 (19.4%) |
| Tramadol | 29527 (10.9%) | 1367 (9.7%) | 462 (9.4%) | 905 (9.8%) |
| Codeine | 17578 (6.5%) | 812 (5.7%) | 276 (5.6%) | 536 (5.8%) |
| Methadone | 11948 (4.4%) | 716 (5.1%) | 249 (5.1%) | 467 (5.1%) |
| Fentanyl | 6141 (2.4%) | 430 (3.0%) | 137 (2.8%) | 293 (3.2%) |
| Hydromorphone | 3439 (1.3%) | 189 (1.3%) | 64 (1.3%) | 125 (1.4%) |
| Buprenorphine | 236 (0.1%) | 13 (0.1%) | 4 (0.1%) | 9 (0.1%) |
| Tapentadol | 103 (0.04%) | 10 (0.1%) | 4 (0.1%) | 6 (0.06%) |
| Oxymorphone | 79 (0.03%) | 6 (0.04%) | 2 (0.04%) | 4 (0.04%) |
| Pentazocine | 47 (0.02%) | 6 (0.04%) | 3 (0.06%) | 3 (0.03%) |
| Butorphanol | 45 (0.02%) | 4 (0.03%) | 1 (0.02%) | 3 (0.03%) |
| Meperidine | 37 (0.01%) | 1 (0.01%) | 0 (0%) | 1 (0.01%) |
| Levorphanol | 20 (0.01%) | 3 (0.02%) | 0 (0%) | 3 (0.03%) |

Values are means and standard deviations (SD) or n and percent (%) except where indicated.

[a] Not included group comprises patients who refused, did not respond before study closeout, or were deceased or unreachable.

**Table 3. Patient characteristics and patient-reported measures by opioid daily dosage category in 6 months before the index date (n = 9074).**

| | Low (<20) | Moderate (20 to <50) | High (50 to <100) | Very high (≥100) | Overall |
|---|---|---|---|---|---|
| | N = 2581 | N = 3855 | N = 1551 | N = 1087 | N = 9074 |
| **Variables from medical records** | | | | | |
| Age in years | 64.7 (0.3) | 63.8 (0.3) | 63.4 (0.4) | 62.5 (0.7) | 63.8 (0.2) |
| Sex, male | 91.7% (1.1%) | 92.8% (0.9%) | 94.1% (1.3%) | 96.4% (0.9%) | 93.1% (0.6%) |
| Race | | | | | |
| White | 75.3% (1.6%) | 76.8% (1.1%) | 80.8% (1.9%) | 85.9% (2.4%) | 78.1% (0.9%) |
| Black | 16.4% (1.4%) | 14.8% (1.0%) | 11.4% (1.5%) | 6.2% (1.1%) | 13.7% (0.7%) |
| Other or unknown | 8.4% (0.7%) | 8.4% (0.7%) | 7.8% (1.3%) | 7.9% (2.3%) | 8.2% (0.5%) |
| Charlson comorbidity score | 1.34 (0.06) | 1.52 (0.05) | 1.42 (0.08) | 1.49 (0.10) | 1.44 (0.03) |
| Pain diagnoses | | | | | |
| Back/spine disorders | 58.7% (1.8%) | 68.5% (1.4%) | 72.5% (2.1%) | 71.3% (3.2%) | 66.9% (1.0%) |
| Neck/spine disorders | 16.7% (1.2%) | 18.6% (1.1%) | 23.6% (1.9%) | 22.4% (3.0%) | 19.5% (0.8%) |
| Osteoarthritis | 27.3% (1.5%) | 29.0% (1.2%) | 29.0% (2.1%) | 25.3% (2.3%) | 28.0% (0.8%) |
| Neuropathy | 16.7% (1.1%) | 19.1% (1.0%) | 22.4% (1.7%) | 22.1% (2.3%) | 19.3% (0.7%) |
| Headache | 7.0% (0.8%) | 6.7% (0.7%) | 6.4% (0.8%) | 6.1% (1.1%) | 6.7% (0.4%) |
| Mental health diagnoses | | | | | |
| Depressive disorder | 26.9% (1.8%) | 26.6% (1.1%) | 33.1% (1.9%) | 34.2% (2.9%) | 28.8% (0.9%) |
| Anxiety disorder | 13.9% (1.1%) | 14.8% (1.0%) | 15.4% (1.5%) | 18.5% (3.0%) | 15.1% (0.7%) |
| PTSD | 21.3% (1.2%) | 22.0% (1.1%) | 26.2% (2.0%) | 22.8% (2.3%) | 22.7% (0.8%) |
| Alcohol use disorder | 6.3% (0.7%) | 6.0% (0.6%) | 5.1% (0.8%) | 3.9% (0.9%) | 5.7% (0.4%) |
| Drug use disorder | 5.2% (0.8%) | 4.9% (0.6%) | 6.5% (0.8%) | 10.8% (1.6%) | 6.0% (0.4%) |
| Opioid formulation | | | | | |
| Any long-acting | 1.2% (0.2%) | 13.2% (1.0%) | 59.3% (2.2%) | 86.9% (2.7%) | 26.8% (0.9%) |
| No long-acting (only short-acting) | 98.8% (0.2%) | 86.8% (1.0%) | 40.7% (2.2%) | 13.1% (2.7%) | 73.2% (0.9%) |
| **Patient-reported variables** | | | | | |
| Past-week average pain severity [0–10] | 6.57 (0.07) | 6.84 (0.05) | 6.77 (0.07) | 6.84 (0.08) | 6.75 (0.04) |
| Number of pain locations [a] [0–10] | 6.55 (0.08) | 6.87 (0.05) | 6.92 (0.09) | 6.94 (0.12) | 6.80 (0.04) |
| General self-rated health | | | | | |
| Very good-excellent | 6.7% (0.8%) | 5.5% (0.8%) | 4.1% (0.7%) | 3.3% (0.9%) | 5.3% (0.5%) |
| Good | 29.4% (1.3%) | 27.0% (1.5%) | 25.8% (2.0%) | 19.8% (3.1%) | 26.6% (0.9%) |
| Fair-poor | 63.8% (1.4%) | 67.6% (1.5%) | 70.1% (2.0%) | 76.9% (3.1%) | 68.1% (0.9%) |
| Effectiveness of pain treatment | | | | | |
| Very good-excellent | 11.7% (1.0%) | 12.3% (1.0%) | 11.9% (1.8%) | 14.0% (2.1%) | 12.3% (0.6%) |
| Good | 32.2% (1.7%) | 30.5% (1.2%) | 31.5% (2.1%) | 34.3% (3.4%) | 31.6% (0.9%) |
| Fair-poor | 56.1% (1.8%) | 57.2% (1.3%) | 56.6% (2.2%) | 51.7% (3.4%) | 56.1% (1.0%) |
| Quality of pain care | | | | | |
| Very good-excellent | 24.7% (1.3%) | 24.0% (1.1%) | 25.8% (2.3%) | 28.1% (3.2%) | 25.0% (0.8%) |
| Good | 29.5% (1.7%) | 30.0% (1.3%) | 30.7% (1.9%) | 27.6% (2.8%) | 29.7% (0.9%) |
| Fair-poor | 45.8% (1.7%) | 46.0% (1.4%) | 43.5% (2.1%) | 44.3% (3.3%) | 45.3% (0.9%) |
| Desire for more or higher dose opioids [c] | | | | | |
| Agree-strongly agree | 35.2% (1.7%) | 37.0% (1.4%) | 40.6% (2.3%) | 35.5% (3.1%) | 37.0% (1.0%) |
| Neutral | 26.6% (1.8%) | 25.2% (1.2%) | 21.6% (1.8%) | 20.6% (1.9%) | 24.4% (0.8%) |
| Disagree-strongly disagree | 38.2% (1.9%) | 37.8% (1.5%) | 37.8% (2.2%) | 43.8% (3.3%) | 38.6% (1.0%) |
| Desire to stop or cut down on opioids [c] | | | | | |
| Agree-strongly agree | 14.3% (1.2%) | 15.9% (1.1%) | 16.6% (1.4%) | 18.5% (2.3%) | 15.9% (0.7%) |
| Neutral | 29.2% (1.4%) | 27.6% (1.2%) | 28.2% (2.1%) | 21.2% (2.1%) | 27.4% (0.8%) |

(*Continued*)

**Table 3.** (Continued)

| | Low (<20) | Moderate (20 to <50) | High (50 to <100) | Very high (≥100) | Overall |
|---|---|---|---|---|---|
| | N = 2581 | N = 3855 | N = 1551 | N = 1087 | N = 9074 |
| Disagree-strongly disagree | 56.5% (1.7%) | 56.6% (1.5%) | 55.1% (2.3%) | 60.3% (2.8%) | 56.7% (1.0%) |

Values are means or percentages and standard errors (SE) weighted to account for study design. Patients with a lost questionnaire (n = 8) or with no other patients in their provider cluster (n = 171) were dropped from analyses.

[a] Count of locations patients reported bothered them "a little" or "a lot" (range 0–10), not including widespread pain." Pilot participants (n = 354) were asked about only 8 locations. Responses were included if at least 7 location items were completed.

[b] 1086 not assessed due to completing a verbal or pilot survey.

[c] 732 not assessed due to completing a verbal survey.

past-week average pain severity was 6.75 (SE 0.04) and the mean number of bothersome pain locations was 6.8 (SE 0.04). Fig 2 shows prevalence of bothersome pain at individual locations. Overall, 68.1% (SE 0.9%) of participants reported their general health was fair-poor. The effectiveness of pain treatment and quality of pain care were rated fair-poor by 56.1% and 45.3%, respectively. Thirty-seven percent reported a desire for more or stronger opioids, whereas 15.9% reported a desire to stop or cut down on opioids.

In the 6 months before the index date, 2579 participants (weighted 26.8%, SE 0.9%) were dispensed at least one long-acting opioid and 6495 (weighted 73.2%, SE 0.9%) were dispensed only short-acting opioids. Of participants who received long-acting opioids, 1829 (weighted 70.9%, SE 1.4%) also received at least one short-acting opioid. For the overall survey cohort, the opioid daily dosage range was 1.6 to 1038.2 mg, the weighted mean was 50.6 mg (SE 1.1), and the weighted median was 30.9 mg (IQR 40.7). Daily dosages were higher among patients who received long-acting opioids (dose range 9.5 to 1038.2 mg; weighted mean 106.2 mg, SE 2.8; weighted median 83.8 mg, IQR 72.9) than among those who did not (dose range 1.6 to 500.0 mg; weighted mean 30.4 mg, SE 0.6; weighted median 25.0 mg, IQR 23.1). Fig 3 illustrates the differing distributions of daily dosages for patients treated with and without long-acting opioids.

## Association of opioid treatment factors with patient-reported outcome measures

We first examined whether treatment with a long-acting opioid modified dosage-outcome relationships. In linear regression models, the interaction term (daily dosage in 10-mg increments x long-acting opioid) was statistically significant in models for BPI-I (p<0.0001) and VR-12 physical (p = 0.0029), and marginally significant for VR-12 mental (p = 0.0466).

Next, separately for each outcome, we examined the effect of treatment with long-acting opioids in models that did not include daily dosage. Compared with participants who received only short-acting opioids, those who received long-acting opioids had worse pain-related function (BPI-I adjusted mean = 6.81, SE = 0.07 vs. 6.40, SE = 0.04; p<0.001) and worse physical health (VR-12 physical score adjusted mean = 22.8, SE = 0.3 vs. 25.4, SE = 0.2; p<0.001), but did not differ on mental health (VR-12 mental score adjusted mean = 38.7, SE = 0.6 vs. 39.6, SE = 0.3; p = 0.168).

Finally, we examined the association of opioid daily dosage (as a continuous variable in 10 mg increments and as a categorical variable) with outcomes in separate models for patients treated with and without long-acting opioids. For patients treated with long-acting opioids, daily dosage was not associated with BPI-I (beta coefficient 0.01, p = 0.0681) or VR-12 physical (beta coefficient -0.04, p = 0.241), but each additional 10 mg was marginally statistically

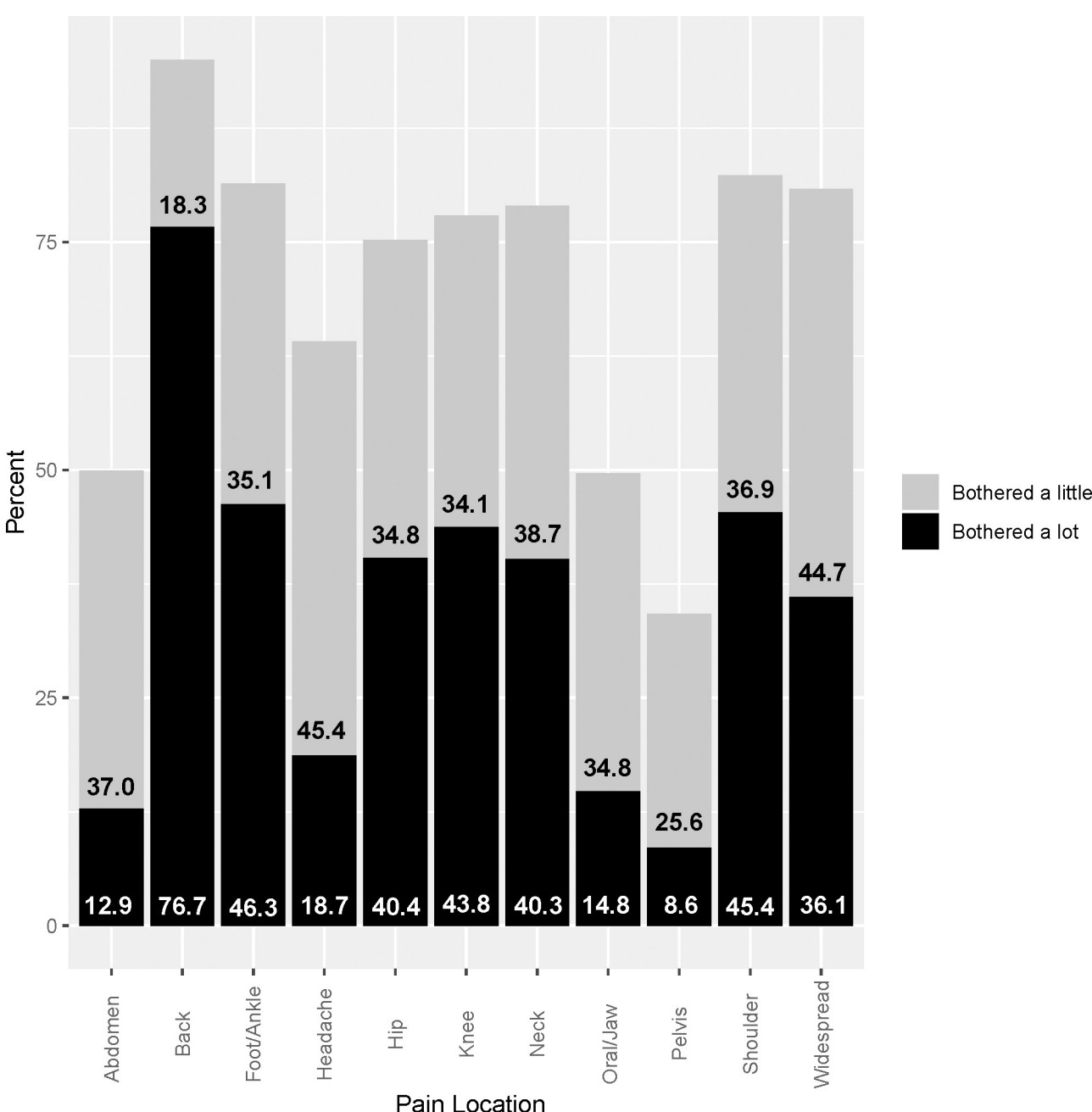

**Fig 2. Bothersome pain locations in the past 6 months among survey cohort participants (n = 9074).** Values are weighted percentages for each response option.

associated with a small decrement in VR-12 mental score (beta coefficient -0.09, p = 0.0236). For patients treated without long-acting opioids, higher daily dosages were significantly associated with worse outcomes; specifically, each additional 10 mg was associated with 0.10-point increase in BPI-I (p<0.0001), 0.41-point decrease in VR-12 physical (p<0.0001), and 0.39-point decrease in VR-12 mental (p = 0.0003). Sensitivity analyses limiting the upper dosage range produced similar results (S5 Table). Table 4 shows outcomes by conventional daily dosage categories for patients with and without long-acting opioids. For participants treated

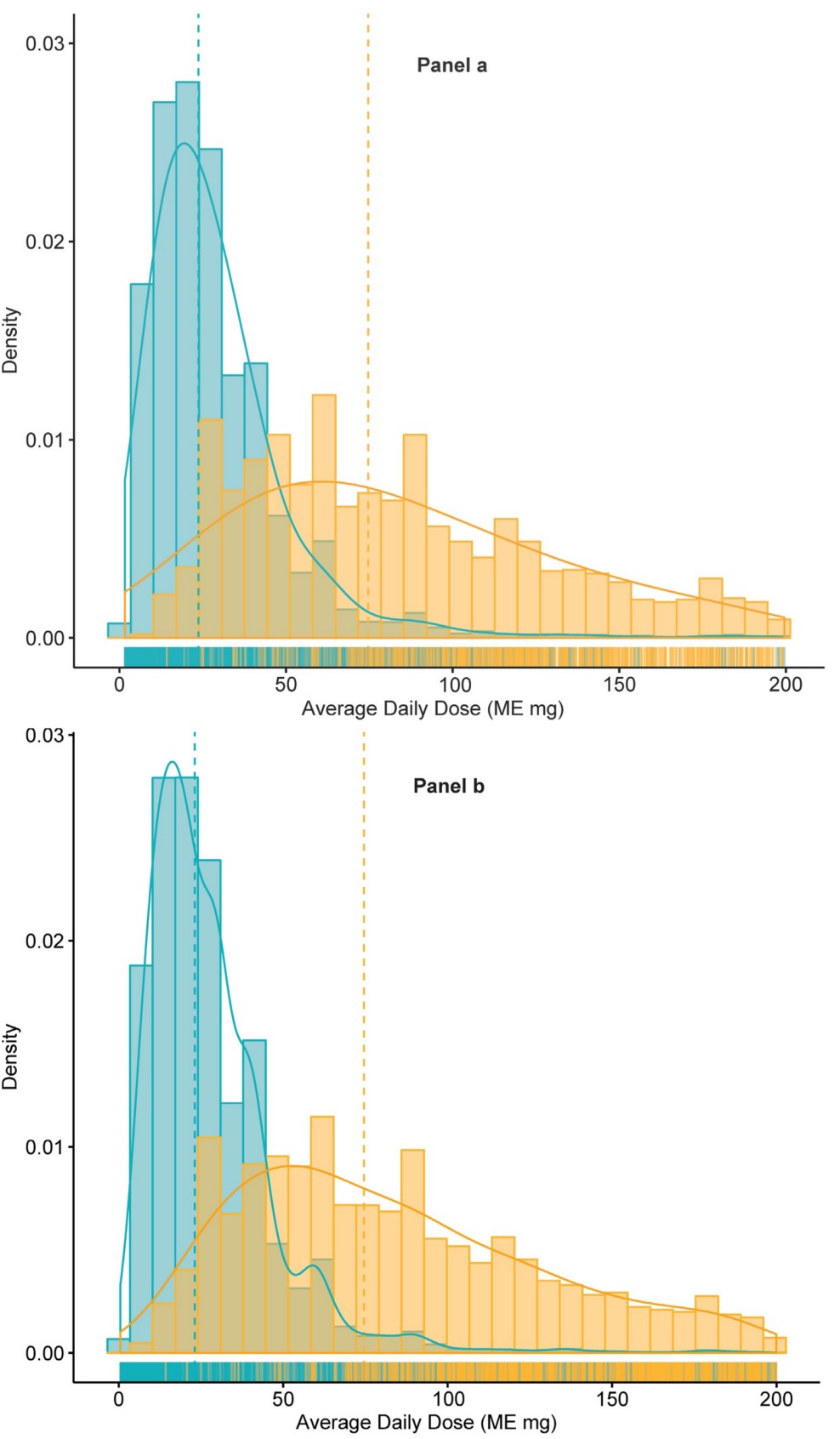

**Fig 3. Density of mean daily dosage by opioid formulation (long-acting opioid versus no long-acting opioid) in 6 months before the index date among a) survey cohort patients and b) eligible patients.** Panel 3a: Survey cohort patients with daily dosage < 200 ME mg (n = 8946). Panel 3b: All eligible patients with daily dosage < 200 ME mg (n = 264,321). Blue = long-acting opioid; yellow = no long-acting opioid. Patients with daily dosage ≥ 200 ME mg are not shown. Panel 3a shows 8946 survey cohort participants with daily dose <200 ME mg (296 with a long-acting opioid and 11 without a long-acting opioid who had daily dosage ≥ 200 ME mg are not shown). Panel 3b shows 264,321 eligible patients with daily dose <200 ME mg (7217 with a long-acting opioid and 354 without a long-acting opioid who had daily dosage ≥ 200 ME mg are not shown).

without long-acting opioids, but not for those treated with long-acting opioids, higher dose categories had significantly worse scores on all three outcomes.

## Discussion

Overall, patients treated with LTOT reported burdensome pain that was multifocal and associated with substantial functional impairment and diminished health. Satisfaction with effectiveness and quality of pain care was low, but most patients were not interested in decreasing opioid use. Indicators of more intensive opioid therapy—higher opioid daily dosages and receipt of long-acting opioids—were associated with worse pain-related function and physical and mental health.

This study had a good survey response rate, which we attribute to use of recommended multiple-contact, multiple-mode survey practices and a brief questionnaire focused on a topic salient to our patient population. In addition, our target population comprised patients who, by definition, were receiving ongoing care from a VA primary care provider. As a result of this ongoing connection, contact information were likely to be relatively up to date and patients may have been more likely to open mail or answer calls from VA researchers. Respondents were somewhat older than non-respondents, as is common in patient surveys. Importantly, we found no evidence of response bias related to pain, opioid dosage, or mental health or substance use diagnoses.

Results indicate patients currently treated with LTOT bear a heavy burden of unrelieved pain and related impairment in function and quality of life. Prior population-based studies have reported associations of opioid use with high levels of pain, functional impairment, and poor quality of life.[24–27] The Pain and Opioids in Treatment (POINT) prospective study of 1500 Australian patients on long-term opioids for chronic pain found these patients faced complex challenges, including multiple pain conditions, poor physical health, and frequent mental health problems.[28] This study confirms these associations in a large US VA clinical population-based cohort.

Most participants in this study reported multiple pain locations and the back was the most common location of bothersome pain. These findings are consistent with previous research, although prior studies are not directly comparable. Chronic back pain is the leading cause of years lived with disability in the US and affects about 75% of US adults who have chronic pain severe enough to limit life activities.[29, 30]

Consistent with published literature, this study found no evidence of better pain control among patients receiving higher intensity opioid therapy. A recent synthesis of evidence from randomized controlled trials (duration 4 weeks to 6 months) found no opioid dose-response relationship for pain or functional outcomes.[31] Prior observational studies have reported statistically significant associations of higher dosage opioid therapy with worse patient-reported outcomes. The Australian POINT study found worse pain interference, higher pain severity, and lower patient-reported relief from medications among patients in higher opioid daily dosage categories.[32] Likewise, an observational study of VA and non-VA patients

**Table 4. Patient-reported outcomes by opioid formulation (long-acting opioid versus no long-acting opioid) and opioid daily dosage category in 6 months before the index date.**

| Outcome [a] | Daily dosage category | | | | | p-value [c] |
|---|---|---|---|---|---|---|
| | Overall | <20 | 20 to <50 | 50 to <100 | ≥100 | |
| **Long-acting opioid** | | | | | | |
| | n = 2579 | n = 53 [b] | n = 594 | n = 938 | n = 994 | |
| **BPI-I [0–10]** | 6.81 (0.07) | 6.17 (0.75) | 6.88 (0.12) | 6.87 (0.11) | 7.05 (0.10) | 0.373 |
| **VR-12 physical** | 22.8 (0.3) | 26.0 (2.6) | 23.5 (0.8) | 22.5 (0.4) | 22.0 (0.4) | 0.175 |
| **VR-12 mental** | 38.7 (0.6) | 33.2 (2.8) | 38.0 (0.8) | 38.2 (0.8) | 37.3 (0.8) | 0.318 |
| **No long-acting opioid (i.e., only short-acting opioids)** | | | | | | |
| | n = 6495 | n = 2528 | n = 3261 | n = 613 | n = 93 [b] | |
| **BPI-I [0–10]** | 6.40 (0.04) | 6.12 (0.07) | 6.44 (0.06) | 6.75 (0.11) | 7.44 (0.19) | <0.001 |
| **VR-12 physical** | 25.4 (0.2) | 26.3 (0.3) | 25.4 (0.3) | 23.5 (0.4) | 20.2 (1.4) | <0.001 |
| **VR-12 mental** | 39.6 (0.3) | 40.9 (0.4) | 39.8 (0.4) | 37.6 (0.8) | 35.5 (2.6) | 0.001 |

BPI-I = Brief Pain Inventory-Interference scale [range 0–10, higher scores indicate worse pain-related function], VR-12 = Veterans RAND 12-item Health Survey [range 0–100, standardized and normed to the US population mean of 50 and SD of 10, higher scores indicate better health-related quality of life]

[a] Missingness varies by outcome. N = 8956 for BPI-I, n = 8777 for VR-12 physical, n = 8763 for VR-12 mental.

[b] Note: few patients were in these categories

[c] P-value for comparison between daily dose categories from Wald test accounting for study design weights and adjusted for age.

found higher pain-related disability and poorer physical function among patients prescribed higher versus lower long-term opioid daily dosages.[33] A prospective observational study of patients initiating new LTOT found those who continued regular opioid use for 12 months had worse pain and functional outcomes than those who minimized or discontinued opioid use.[34] We are not aware of prior studies that examined outcomes among patients treated with versus without long-acting opioids; however, one prior study found similar levels of pain —as well as higher daily dosages and more concerns about opioid dependence—among patients who took opioids on a fixed schedule (as is recommended with long-acting opioids) compared with those who took opioids on an as-needed basis.[35]

We examined relationships among opioid formulation and dosage in more detail than prior studies and found the distribution of opioid daily dosage differed substantially between patients treated with and without long-acting opioids: patients with long-acting opioids received much higher daily dosages. Further, we identified an interesting two-part finding related to patient-reported outcome measures; first, patients treated with long-acting opioids had worse outcomes overall than those treated without long-acting opioids, and second, the association of higher daily dosage with worse outcomes held only for patients treated without long-acting opioids. Although pain-related function and quality of life were generally poor in this cohort, patients with the most favorable outcomes were those who received lower dosage short-acting-only opioid regimens and who were therefore unlikely to have "around the clock" opioid coverage. Hypothetically, these results could be due to selection of more intensive opioid regimens for more ill patients, adverse effects of higher intensity opioid regimens (e.g., opioid-induced hyperalgesia), a dose ceiling effect on opioid analgesia, or to some combination of these causes.[36]

In this study, only a minority of patients reported a desire to reduce opioid use, despite widespread dissatisfaction with results of current pain management. Across dosage categories, patients more often desired an increase than a decrease in opioid treatment intensity. Qualitative studies have described fears and beliefs that potentially underlie these patient preferences, including pessimism about non-opioid therapies, perceptions that opioids are more effective

than other pain medications, and fears of uncontrolled pain, withdrawal, or abandonment. [37–40]

The major strengths of this study include the large national sample, good survey response rate, and linkage to high-quality EMR and pharmacy dispensing data. This study also has limitations. First, the cohort is not representative of all US primary care patients on long-term opioid therapy. The VA patient population differs in demographics and life experience and may have a higher prevalence of medical and psychiatric conditions than other US patient populations. Likewise, opioid prescribing in VA may differ from prescribing in non-VA settings.[41] Second, opioid treatment data are from VA outpatient pharmacy records only; opioid prescriptions from non-VA prescribers and pharmacies were not captured. Third, our data sources (both questionnaires and administrative data) have limitations. For example, whereas self-report is the best approach for assessing pain severity and administrative data are highly accurate for quantifying medication dispensing, neither approach accurately identifies the presence of clinically diagnosed conditions such as osteoarthritis and opioid use disorder. Further, self-report measures are subject to problems such as recall bias and social desirability bias. Fourth, we cannot infer directionality of associations in this report of cross-sectional analyses. The EPOCH study is collecting longitudinal data, including annual follow-up surveys. Planned analyses of longitudinal data will examine effects of changes in treatment on outcomes.

## Supporting information

**S1 Table. Opioid formulations and dosage conversion factors.**
(DOCX)

**S2 Table. Patient exclusion criteria.**
(DOCX)

**S3 Table. Pain and mental health diagnosis codes categories.**
(DOCX)

**S4 Table. Analysis of response among eligible patients selected for invitation (n = 13,976).**
(DOCX)

**S5 Table. Relationship of opioid daily dosage in 10 mg increments with outcome measures in patients treated with and without long-acting opioids.**
(DOCX)

## Acknowledgments

- We thank the veteran participants in the study and the members of the research team, including Ann Bangerter and Andrea Cutting (data team) and Erin Amundson, Ruth Balk, Cody Bassett, Abigail Klein, David Leverty, Erin Linden, and Derek Vang (research assistants).

- The views expressed in this article are those of the authors and do not necessarily represent the views of the US Government or Department of Veterans Affairs.

## Author Contributions

**Conceptualization:** Erin E. Krebs, Brian C. Martinson, Siamak Noorbaloochi.

**Data curation:** Barbara Clothier, Sean Nugent.

**Formal analysis:** Barbara Clothier, Sean Nugent, Siamak Noorbaloochi.

**Funding acquisition:** Erin E. Krebs.

**Investigation:** Erin E. Krebs, Barbara Clothier, Agnes C. Jensen, Brian C. Martinson, Elizabeth S. Goldsmith, Melvin T. Donaldson, Joseph W. Frank, Siamak Noorbaloochi.

**Methodology:** Erin E. Krebs, Barbara Clothier, Sean Nugent, Brian C. Martinson, Elizabeth S. Goldsmith, Melvin T. Donaldson, Joseph W. Frank, Siamak Noorbaloochi.

**Project administration:** Erin E. Krebs, Sean Nugent, Agnes C. Jensen, Indulis Rutks.

**Resources:** Agnes C. Jensen, Indulis Rutks.

**Supervision:** Erin E. Krebs, Agnes C. Jensen, Siamak Noorbaloochi.

**Validation:** Barbara Clothier, Sean Nugent, Agnes C. Jensen, Siamak Noorbaloochi.

**Writing – original draft:** Erin E. Krebs.

**Writing – review & editing:** Erin E. Krebs, Barbara Clothier, Sean Nugent, Agnes C. Jensen, Brian C. Martinson, Elizabeth S. Goldsmith, Melvin T. Donaldson, Joseph W. Frank, Indulis Rutks, Siamak Noorbaloochi.

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
