## [Decision Letter · Decision Letter 0]

19 Dec 2019

PONE-D-19-31436

The Evaluating Prescription Opioid Changes in Veterans (EPOCH) study: design, survey response, and baseline characteristics

PLOS ONE

Dear Dr Krebs,

Thank you for submitting your manuscript to PLOS ONE. After careful consideration, we feel that it has merit but does not fully meet PLOS ONE’s publication criteria as it currently stands. Therefore, we invite you to submit a revised version of the manuscript that addresses the points raised during the review process.

We would appreciate receiving your revised manuscript by Feb 02 2020 11:59PM. To enhance the reproducibility of your results, we recommend that if applicable you deposit your laboratory protocols in protocols.io, where a protocol can be assigned its own identifier (DOI) such that it can be cited independently in the future. For instructions see: http://journals.plos.org/plosone/s/submission-guidelines#loc-laboratory-protocols

We look forward to receiving your revised manuscript.

Kind regards,

Yan Li

Academic Editor

PLOS ONE

Journal Requirements:

2. Please address the following:

- Please ensure you have thoroughly discussed any potential limitations of this study within the Discussion section, including the potential bias introduced by using self-reported data.

- Please further describe how the "minimum panel size criteria" was calculated.

Thank you for your attention to these queries.

3.

In your Data Availability statement, you have not specified where the minimal data set underlying the results described in your manuscript can be found. PLOS defines a study's minimal data set as the underlying data used to reach the conclusions drawn in the manuscript and any additional data required to replicate the reported study findings in their entirety. All PLOS journals require that the minimal data set be made fully available. For more information about our data policy, please see http://journals.plos.org/plosone/s/data-availability.

4. Please upload a copy of Figure 3, to which you refer in your text on page xx. If the figure is no longer to be included as part of the submission please remove all reference to it within the text.

Reviewers' comments:

Reviewer's Responses to Questions

**Comments to the Author**

1. Is the manuscript technically sound, and do the data support the conclusions?

Reviewer #1: Yes

Reviewer #2: Yes

2. Has the statistical analysis been performed appropriately and rigorously? 

Reviewer #1: Yes

Reviewer #2: Yes

3. Have the authors made all data underlying the findings in their manuscript fully available?

Reviewer #1: Yes

Reviewer #2: Yes

4. Is the manuscript presented in an intelligible fashion and written in standard English?

Reviewer #1: Yes

Reviewer #2: Yes

5. Review Comments to the Author

Reviewer #1: This study about EPOCH has provided valuable information about the outcome of reduced/discontinued prescribed opioid among veterans and has set an example to conduct related studies. The survey is carefully designed and data is rationally analyzed. The description about results is precise as well. I personally enjoyed reading this manuscript. Some minor concerns are listed below:

1. In table 1, it shows back/spine disorder is the most commonly pain type in the survey. Is there special reason leading to massive back/spine injuries among veterans?

2. This survey has a very good response rate. What would the author think is the most important contributor?

3. I am personally very curious about what can be a potential alternative for opioid?

Reviewer #2: Thank you for the opportunity to review this manuscript, this paper is technically sound.

Please organize the tables, it's way too busy, very hard to went through, better split to 2-3 different tables,

If Fig 1 is using the table version for presenting, please rename as Table

6. PLOS authors have the option to publish the peer review history of their article (what does this mean?). If published, this will include your full peer review and any attached files.

Reviewer #1: No

Reviewer #2: No

---

## [Author Response · Author response to Decision Letter 0]

15 Feb 2020

Editorial comments:

We have applied style requirements, including those for file names.

2. Please ensure you have thoroughly discussed any potential limitations of this study within the Discussion section, including the potential bias introduced by using self-reported data.

We added limitations related to self-report and administrative data (page 22, lines 486-491).

3. Please further describe how the "minimum panel size criteria" was calculated. 

We added details about our approach to identifying primary care providers and applying minimum panel size criteria. To improve clarity for readers, we put this information in a new methods subsection, “survey sample selection.” (pages 6-7, lines 145-166)

4. Please upload a copy of Figure 3, to which you refer in your text. 

Figure 3 is included with the submission. 

5. Please include captions for your Supporting Information files at the end of your manuscript, and update any in-text citations to match accordingly. 

Supporting information captions are now included at the end of the manuscript and in-text citations are updated to match. (pages 28-29, lines 631-637)

Reviewer #1 comments: 

1. In table 1, it shows back/spine disorder is the most commonly pain type in the survey. Is there special reason leading to massive back/spine injuries among veterans?

We added information about back pain prevalence to the discussion. (age 20, lines 434-438)

2. This survey has a very good response rate. What would the author think is the most important contributor?

We used multiple methods that have been found to improve response rates or that we hoped would help, so we can’t disentangle the most important reasons for our success. I added to the discussion an additional contributor—the ongoing clinical relationship study patients had with VA clinics. (page 19, lines 418-421)

3. I am personally very curious about what can be a potential alternative for opioid?

Although care should be individualized, guidelines recommend a variety of other medications, exercise therapies, psychological therapies such as cognitive behavioral therapy, manual treatments such as spinal manipulation, and mind-body approaches such as yoga. 

Reviewer #2 comments:

1. Please organize the tables, it's way too busy, very hard to went through, better split to 2-3 different tables.

We split Tables 1 into two tables, separating patient characteristics from opioid treatment received. (Pages 12 and 13) I am unsure of a better way to present data in the other tables but am open to specific suggestions you may have. 

2. If Fig 1 is using the table version for presenting, please rename as Table

Fig 1 is the study flow diagram and is not in table format.

---

## [Editor Report · Decision Letter 1]

9 Mar 2020

The Evaluating Prescription Opioid Changes in Veterans (EPOCH) study: design, survey response, and baseline characteristics

PONE-D-19-31436R1

Dear Dr. Krebs,

We are pleased to inform you that your manuscript has been judged scientifically suitable for publication and will be formally accepted for publication once it complies with all outstanding technical requirements.

With kind regards,

Yan Li

Academic Editor

PLOS ONE
---

## [Editor Report · Acceptance letter]

10 Apr 2020

PONE-D-19-31436R1 

The Evaluating Prescription Opioid Changes in Veterans (EPOCH) study: design, survey response, and baseline characteristics 

Dear Dr. Krebs:

I am pleased to inform you that your manuscript has been deemed suitable for publication in PLOS ONE. Congratulations! Your manuscript is now with our production department. 

With kind regards,

on behalf of

Dr. Yan Li 

Academic Editor

PLOS ONE